# In Search of Molecular Markers for Cerebellar Neurons

**DOI:** 10.3390/ijms22041850

**Published:** 2021-02-12

**Authors:** Wing Yip Tam, Xia Wang, Andy S. K. Cheng, Kwok-Kuen Cheung

**Affiliations:** 1University Research Facility in Behavioral and Systems Neuroscience, The Hong Kong Polytechnic University, Hong Kong, China; 2Department of Anatomy and Histology, School of Basic Medical Sciences, Shenzhen University Health Science Centre, Shenzhen 518000, China; xia.wang@szu.edu.cn; 3Department of Rehabilitation Sciences, The Hong Kong Polytechnic University, Hong Kong, China; andy.cheng@polyu.edu.hk

**Keywords:** cerebellum, genetics, single-cell transcriptome, laser-capture microdissection, next-generation sequencing, neuronal marker

## Abstract

The cerebellum, the region of the brain primarily responsible for motor coordination and balance, also contributes to non-motor functions, such as cognition, speech, and language comprehension. Maldevelopment and dysfunction of the cerebellum lead to cerebellar ataxia and may even be associated with autism, depression, and cognitive deficits. Hence, normal development of the cerebellum and its neuronal circuitry is critical for the cerebellum to function properly. Although nine major types of cerebellar neurons have been identified in the cerebellar cortex to date, the exact functions of each type are not fully understood due to a lack of cell-specific markers in neurons that renders cell-specific labeling and functional study by genetic manipulation unfeasible. The availability of cell-specific markers is thus vital for understanding the role of each neuronal type in the cerebellum and for elucidating the interactions between cell types within both the developing and mature cerebellum. This review discusses various technical approaches and recent progress in the search for cell-specific markers for cerebellar neurons.

## 1. Introduction

The cerebellum is the primary brain region responsible for motor coordination and balance. Recent studies indicate that the cerebellum also contributes to non-motor functions, such as cognition, vision, audition, and language production and comprehension [1,2]. Many cerebellar neurons are required to meet the demand for performing these functions; indeed, the cerebellum contains more than 50% of the total number of neurons in the central nervous system (CNS), packed within 10% of the total brain volume [3]. Interestingly, despite the cerebellum’s high cellular density, cerebellar circuitry is relatively simple compared to that of the cerebral cortex, as the former comprises only a few neuronal cell types in the cerebellar cortex [4,5].

### 1.1. The Cerebellar Cortex

The cerebellar cortex consists of three layers: the outermost molecular layer (ML), the innermost granular layer (GL), and the intermediate Purkinje cell layer (PCL; Figure 1). At the ML, extensive dendritic arborization of Purkinje cells (PCs) arranged in the parasagittal plane forms perpendicular synaptic connections with the axons (parallel fibers) of granule cells (GCs). Moreover, two types of GABAergic interneurons—stellate and basket cells—serve to supplement inhibitory inputs to PCs [6,7,8]. At the PCL, where PCs are located, candelabrum cells (another type of interneuron) are found between PC somata [9]. Meanwhile, GCs and four types of interneurons—Golgi cells, globular cells, Lugaro cells, and unipolar brush cells—are located in the innermost GL. PCs relay all inputs from parallel fibers and interneurons and send the final output to the deep cerebellar nuclei, which further direct the signals to the thalamus and motor system to control body movement fine-tuning [10,11,12]. Furthermore, two major extra-cerebellar afferents are projected to the cerebellar cortex: mossy fiber and climbing fiber. Whereas mossy fibers emerge from the pre-cerebellum nuclei (brainstem reticular and pontine nuclei) and are connected to GCs and deep cerebellar nuclei, climbing fibers originate from the inferior olive and connect to PC dendrites [13,14] (Figure 1).

### 1.2. Development of the Mouse Cerebellum

Mouse cerebellum development involves multiple steps [17]. During embryonic development, a molecular signaling center (isthmic organizer) is established at the dorsal region of rhombomere 1 between embryonic days (E) 8.5 and 9. The isthmic organizer specifies the mid/hindbrain boundary, and the cerebellar primordium develops from the hindbrain region, beginning at E10 [18]. Two germinative zones are established in the cerebellar primordium: the ventricular zone and the rhombic lip. Whereas PCs and other GABAergic interneurons (e.g., Golgi cells, globular cells, Lugaro cells, basket cells, stellate cells, and candelabrum cells) are derived from the ventricular zone, GCs—the most abundant neurons in the cerebellum—and another type of glutamatergic interneuron unipolar brush cell are derived from the rhombic lip [13,19,20]. From E10.5 onward, postmitotic PCs emigrate from the ventricular zone, migrating toward the pial surface, where the PCs colonize, differentiate, and give rise to the PC layers of the cerebellum [17]. GCs emigrate from the rhombic lip and migrate along the pial surface to form the external GL. After birth, the GCs in the external GL migrate inwardly to form the internal GL, while PC layers align to form a single PCL. Dendritic arborization of PCs expands and forms synapses with parallel fibers of GCs at the ML. The foliation of the cerebellum is also initiated at the perinatal stage, and cerebellar size expands rapidly. Eventually, the cerebellar circuitry reaches maturity around postnatal day (P) 21 [4,14,17].

### 1.3. Importance of Neuronal-Specific Markers in Cerebellum Research

Maldevelopment of the cerebellum and dysfunction of the cerebellar circuitry lead to cerebellar ataxia, a neurological disease characterized in humans by body tremors, impaired body balance, and gait instability [21]. Defective cerebellum may also be associated with other non-motor conditions, such as depression and cognitive deficits [22,23,24,25,26,27,28]. Essentially, the ability to characterize each neuronal cell type in the cerebellum is crucial for understanding the pathogenesis of cerebellar disorders. However, with the exception of PCs and GCs, cell-specific markers for most (if not all) cerebellar interneurons remain elusive. This deficit may produce a bottleneck in cerebellum research, as cell-specific manipulations such as cell labeling and genetic knockout/knockin (KO/KI) are unfeasible without genetic markers, hindering the examination of the functions of particular cell types.

The importance of cell-specific markers in cerebellum research is best illustrated by studies concerning PCs in which several PC-specific markers are available. For example, Purkinje cell protein 2 (*Pcp2*/*L7*) is a protein specifically expressed in PCs, and its gene promoter has been utilized for driving PC-specific transgene expression [29]. One *Pcp2*-Cre transgenic mouse line expresses Cre recombinase specifically in PCs, beginning at P6, and is particularly useful in inactivating the floxed gene during cerebellar circuitry establishment at the postnatal stage [30]. The *Pcp2* promoter has also been used to drive disease–gene expression (e.g., expression of *Ataxin1* with CAG expansion in spinocerebellar ataxia type 1 mouse model) for pathological study [31]. Another PC-specific marker calbindin (*Calb1*) has been widely adopted in immunohistochemical studies of the cerebellum since the late 1980s [32,33]. Furthermore, the availability of cell-specific markers is critical in establishing stem-cell therapy, as the markers aid in consolidating the differentiation program of stem cells or induced pluripotent stem cells (iPSCs) to a specific cell type, enabling the replenishment of adult neurons [34,35,36].

## 2. Approaches to Discovering Molecular Markers of Cerebellar Neurons

In order to search for cerebellar neuronal markers in mice, different research groups have adopted various approaches such as classical genetic approaches (e.g., forward and reverse genetics), global screening (e.g., the Allen Brain Atlas, GENSAT) and “omic” profiling (e.g., transcriptome).

### 2.1. Forward Genetics—From Phenotype to Genotype

Laboratory mice have been used in genetic studies for over a century, and mouse lines—including those with spontaneous genetic mutations—are maintained by inbreeding [37]. Before the advent of advanced molecular technology, many mouse models were named according to their phenotypes instead of their underlying causative genes, which had yet to be discovered. For example, *Staggerer* was named for its staggering gait and body tremor [38], and *Lurcher* was a mutant characterized by “jerky up and down movement” and wobbly gait [39,40]. When positional cloning and DNA sequencing techniques became available, the causative genes were identified in these mutant mice—such as *Rora* in *Staggerer* mice [41] and *Grid2* in *Lurcher* mice [42]. *Lurcher* mice are known to undergo early cerebellar degeneration from the second postnatal week onward, and massive PC death has been observed in the cerebellum as early as P5, revealing that *Grid2* is essential for PC survival [43]. *Grid2* encodes the δ2 glutamate receptor subunit, which is highly localized at the dendritic spines of PCs. Therefore, *Grid2* is considered both a PC-specific marker [44] and an essential target for studying the PC–GC connection [45].

Apart from natural mutations, random mutagenesis induced by chemicals (e.g., N-ethyl-N-nitrosourea (ENU)) or by genetic methods (e.g., transposon and gene trap) has been used to generate mutant mice for global screenings of disease phenotypes [46,47,48,49,50,51]. For example, *Pingu* (a novel mutant with cerebellar ataxia phenotype) was generated by ENU-induced missense mutation (I402T) in the voltage-gated potassium channel *Kcna2* [52]. *Kcna2* is expressed in the axon terminals (pinceau) of basket cells that form synaptic connections at the axon initial segment (AIS) of PCs [52,53,54,55,56]. These studies suggest that a normal connection between basket cell axons and the AIS of PCs is required for motor coordination and body balance.

Another example is the ENU-induced missense mutation of *Af4* in the robotic mouse, which displays a jerky, ataxic gait [57]. *Af4* was initially known for its involvement in the development of acute lymphoblast leukemia [58]. Examination of the robotic mouse demonstrates that the *Af4* mutation also contributes to PC degeneration. *Af4* is expressed in postnatal PCs; the expression peaks at P14 but becomes undetectable at P56 [59]. *Af4*, which has been suggested to be an IGF1-signaling pathway regulator, is critical for the survival of PCs during early postnatal development. Although both *Kcna2* and *Af4* are essential for the maintenance and functions of basket cells and PCs, whether the two genes and their gene promoters can be used for cell-specific labeling requires further characterization.

Random mutations generated by genetic methods are particularly useful for locating cell-specific gene promoter/enhancer regions that can be utilized for transgene expression. In a large-scale screening of enhancer trapping with the *piggyBac* transposon system, more than 200 mouse lines were generated, a number of which have shown cell-type-specific expression of the reporter gene in the CNS [60]. In the cerebellum, some mouse lines demonstrate cell-specific expression of the reporter gene in PCs, GCs, basket cells, and Lugaro cells, making them applicable in cell-specific labeling experiments. Moreover, the *piggyBac* system enables identification of the insertion site, allowing the gene promoter/enhancer region with the transposon insertion to be further characterized for the generation of cell-specific driver mouse strains [60].

Recent advances in genome editing technology have led to the adoption of CRISPR as a forward-genetic tool [61,62]. How CRISPR creates precise target-specific mutations has been reviewed elsewhere [62,63,64], and as such, is not discussed here. Importantly, high-throughput, genome-wide screening of mutants for certain phenotypes can be achieved via in silico designed sgRNA pools, and the process of mutant generation is much faster than conventional gene targeting [61,62].

### 2.2. Reverse Genetics—From Genotype to Phenotype

In contrast to forward genetics, the traditional reverse genetic approach begins with a specific target gene to be mutated. The resultant phenotype is then observed in the mutant animal. The classical methods include replacing the coding sequence of the gene of interest with an antibiotic gene (as a selection marker during embryonic stem cell selection; i.e., gene targeting) and site-directed mutagenesis to produce the desired gene mutation [65]. However, the shortcoming of traditional gene targeting is early lethality resulting from inactivation of essential developmental genes, hindering further study in postnatal and adult stages. For example, *Lhx1* and *Lhx5* are homeobox transcription factors critical for early differentiation of PCs during the embryonic stage [66]. Traditional KO of *Lhx1* results in embryonic lethality, and that of *Lhx5* results in postnatal lethality [67,68]. Therefore, studying the functions of *Lhx1* and *Lhx5* in PCs during postnatal cerebellum development becomes impossible.

Circumventing embryonic/postnatal lethality has involved establishing a conditional KO strategy using the Cre-*loxP* system [69]. In this system, two *loxP* segments are inserted to flank the target gene sequence, which can be either the transcription start site (to inhibit gene expression) or an important functional domain (to inactivate specific protein function). Upon the binding of Cre recombinase to the *loxP* sites, the flanked DNA fragment is excised to mutate the target gene. Achieving tissue- or cell-specific conditional KO requires a Cre-driver mouse, which expresses Cre recombinase in a tissue- or cell-specific manner. For example, crossing the aforementioned *Pcp2*-Cre transgenic mouse lines [30] with the *Lhx1-Lhx5* double-conditional KO mouse reveals that *Lhx1* and *Lhx5* are essential for PC dendritogenesis during postnatal cerebellum development [70]. Overall, the success of cell-specific KO or KI is highly dependent on having a well-characterized Cre-driver mouse [71]. Again, CRISPR technology provides a faster way to generate target-specific KO or KI mouse models. Consequently, CRISPR can also be applied as a reverse-genetic tool [61].

### 2.3. Global Screening

As mentioned, a well-characterized gene promoter is critical for cell-specific labeling and conditional KO or KI experimentation. Hence, an essential step is to find out cell-specific gene expression in the cerebellum to screen out potential gene promoters. Various global screening methods have been attempted to identify cerebellar neuronal markers. A traditional way was tissue-specific cDNA library screening, in which mRNAs extracted from cerebellum were used for cDNA library preparation for initial screening. Genes were then identified by DNA sequencing and in situ hybridization (and immunohistochemical staining for gene products) was performed to obtain spatiotemporal gene expression pattern. *Pcp2*, for example, was one of the cDNA clones identified in the *Purkinje cell degeneration* (*pcd*) mouse [72]. Later, Rong et al. (2004) conducted a microarray study to compare the differential gene expression profiles between wild-type and *pcd^3J^* cerebellum at four months of age, the time when PCs are absent in the *pcd^3J^* cerebellum due to degeneration [73]. Several well-known PC markers (e.g., *Pcp2*, *Calb1*, and *Grid2*), together with other uncharacterized genes, appear in the microarray dataset, supporting the efficient application of microarray profiling in finding potential cell markers in mutant mice. Further examination of these uncharacterized genes may reveal novel PC markers [73].

Another study focuses on the expression of transcription factors in the developing cerebellum shown by in situ hybridization [74]. The researchers in this study identified 24 transcription factors expressed in a cell-type-specific and developmental stage-specific manner. Five transcription factors (*Esrrb*, *Nr2F2*, *Foxp2*, *Foxp4*, and *Rora*) were specifically expressed in PCs from P0 to P30. That said, *Rora* has also been shown to be expressed in interneurons at the molecular layer [75,76]. Moreover, four transcription factors (*Zic1*, *Neurod1*, *Etv1*, and *Nfia*) are specifically expressed in GCs from P0 to adulthood [74]. This study illustrates how global screening of mRNA expression can facilitate the discovery of both cell-specific and stage-specific novel cell markers.

The Allen Mouse Brain Atlas is an open online resource providing extensive and systematic high-resolution in situ hybridization data [77,78], including the mRNA expression pattern of transcription factors as well as that of other protein-coding genes. This database allows users to search for the expression of specific genes in the different anatomical structures of the mouse brain at various developmental stages. For example, the database can display genes that are abundantly expressed in the PCL or GL, narrowing the list of genes expressed specifically in PCs and GCs, respectively. 

GENSAT (Gene Expression Nervous System Atlas) is another open resource that aims to build a public database with gene expression profiles in the developing and adult CNS of the mouse [79,80]. Numerous BAC-EGFP transgenic mice have been produced by replacing the coding sequence with Enhanced Green Fluorescent Protein (EGFP) and then examining the expression pattern of EGFP. Various transgenic mice show distinct spatiotemporal patterns of EGFP expression, and these patterns can be searched in the GENSAT database according to parameters such as brain structure, expression level and pattern, and mouse age. Apart from EGFP reporter mice, BAC Cre-driver mouse lines have also been created [81,82]. Despite the GENSAT *Egr1*-EGFP transgenic mice displaying EGFP expression deviated from the endogenous *Egr1* expression pattern [83], the GENSAT project is a valuable resource for EGFP reporter and Cre-driver mouse lines.

Recently, expression data derived from transcriptomic studies, in situ hybridization, and immunofluorescence staining have been compiled in the Human Protein Atlas’s Brain Atlas [84,85]. This open-access database provides an integrative, comprehensive platform for molecular mapping of gene expression at the mRNA and protein levels for different brain regions or different cell types. In all, these open resources enable researchers to search for potential cell-specific markers using specific parameters. 

### 2.4. Laser-Capture Microdissection, qPCR, and Next-Generation Sequencing

One technical challenge in discovering cell-specific markers is the heterogeneity of tissue consisting of multiple cell types. Conventional mRNA/protein extraction from the whole cerebellum inevitably contains a catalog of mRNA/proteins from mixed populations of neurons and glial cells. Laser-capture microdissection (LCM) provides a convenient method for isolating specific types of cells from histological sections. The general LCM procedure involves selecting particular cells from a histological section under an LCM microscope. Next, a laser beam is excited along the boundary of the target cells. The laser-dissected cells are then captured in a collection tube, and RNA/protein extraction is performed [86]. The LCM method has proven useful for amplifying partially degraded RNA using qPCR [87] and studying miRNAs expression in PCs [88] and formalin-fixed-paraffin-embedded (FFPE) samples from postmortem human brain samples [89,90]. LCM enables dissecting cell layers—for example, external GL tissues from developing cerebellum at E13, E15, and E18—along with studying gene expression profiles of GC precursors [91]. Well-known GC precursor markers, such as *Atoh1* and *Klf4*, can be captured from the external GL LCM samples, suggesting that LCM can effectively enable isolating target cells with minimal contamination of other surrounding cell types [91].

Another study attempted to examine the expression of G protein-gated potassium channels, or potassium inwardly-rectifying channels (GIRKs), in PCs, basket cells, stellate cells, and GCs isolated by LCM [92]. Despite the small size of the cell bodies of the basket and stellate cells and the possibility of contamination from surrounding cells, as validated by other expression studies such as RT-PCR and immunostaining, distinctive expression patterns of the GIRK channels are observable among these cerebellar neurons [92]. Nevertheless, unipolar brush cells cannot be isolated using LCM, as they are surrounded by numerous GCs, implying a limitation of LCM regarding dissecting a distinctive cell from large cell populations [92]. Furthermore, although LCM is a convenient method for extracting mRNA from selected cells, the quality of the extracted mRNA may be poor compared to live-tissue samples, as mRNAs from FFPE samples may be fragmented, especially in the case of aged samples [93]. Moreover, the quantity of RNA retrieved from LCM samples is low [91], potentially rendering the mRNA expression profile incomprehensive. 

### 2.5. Single-Cell Transcriptome

Other methodologies have likewise been developed to ensure good quality mRNAs for comprehensive transcriptome profiling. For example, recent developments of next-generation sequencing technology have popularized single-cell transcriptome analysis. This technology’s general procedures involve dissecting cerebellar tissue, which is then dissociated into single cells. After these cells have been captured, RNA-seq libraries are constructed, and sequencing data are obtained and analyzed using statistical algorithms. Cell lineages are clustered according to gene expression profiles.

Several studies have adopted this approach in searching for neuronal markers in the cerebellum [20,94,95,96,97]. For example, single-cell RNA-seq is performed to study the gene expression of the developing cerebellum at eight prenatal time points (daily sampling at E10 to E17) and four postnatal time points (P0, P4, P7, P10) [94]. The data help to molecularly delineate cerebellar cell types, such as neurons and glial cells, and illustrate essential factors that regulate glutamatergic neuron specification [94].

In another study focusing on postnatal cerebellum development, cerebellar tissues dissected from C57BL/6J mice at P0 and P8 were dissociated into single cells, and single-cell RNA-seq was performed. Among the >20,000 cells analyzed, eight main cell clusters were identified [95]. Several novel PC-specific marker genes were found, such as adenylate cyclase type 1 (*Adcy1*), inositol-trisphosphate 3-kinase A (*Itpka*), and cold shock domain-containing protein C2 (*Csdc2*). Moreover, contactin 2 (*Cntn2*) and tubulin beta 3 class III (*Tubb3*) were also shown to be exclusively expressed in mature GCs, indicating their potential as GC markers. However, given the lack of clear interneuron-specific markers that could be inferred for gene clustering analyses, only general groupings of GABAergic interneurons (i.e., *Ptf1a*/*Ascl1* positive) and rhombic-lip–derived interneurons (i.e., *Atoh1*/*Reelin* positive) could be established. Therefore, genes expressed in interneuron clusters require further examination to search for specific markers for the interneuron subtypes [95].

One drawback of the gene clustering method is its reliance on already identified cell-lineage markers. Because PCs and GCs have several known markers, they can be clustered with high confidence; however, since interneuron-specific markers are not available, gene clustering to identify a particular interneuron type is impossible, and only general groups of interneurons (e.g., Ptf1a^+ve^ or Atoh1^+ve^ interneurons) can be delineated [95]. Hence, further characterizations of expression patterns by immunostaining and in situ hybridization of the genes in these interneuron clusters are still needed for finding cell-type-specific markers.

## 3. Concluding Remarks and Future Perspectives

Although many reviews concern the development of the cerebellum and cerebellar circuitry [3,4,5,17], this review has focused on different technical approaches to discovering potential cerebellar neuronal markers, which are essential for cerebellum research in that they enable cell-specific labeling and genetic manipulation. Two major types of cerebellar neurons—PCs and GCs—have several well-characterized, cell-specific markers. In contrast, cell markers for other interneurons are still lacking. Some examples of potential cerebellar neuronal markers are listed in Table 1. Each of the approaches discussed in this review has advantages and disadvantages, making no single method perfect for finding cell-specific markers. However, integrating different technical approaches can facilitate identifying cell-specific markers for each of the cerebellar neurons, enabling a greater understanding of the functions of each neuronal type and clarifying the general role of the cerebellum.

## Figures and Tables

**Figure 1 ijms-22-01850-f001:**
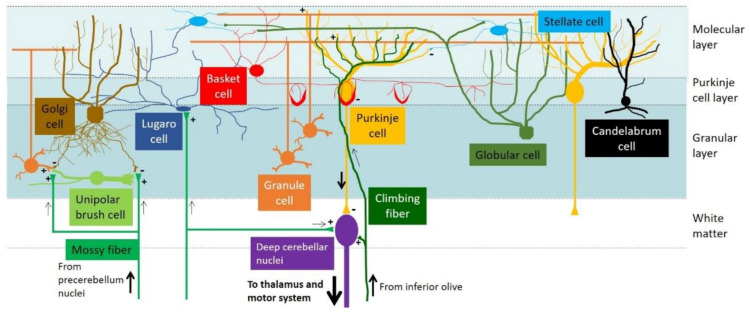
Cytoarchitecture and neuronal circuitry of the mouse cerebellar cortex. Dendrites of Purkinje cells (PCs) form synaptic connections with the parallel fibers of granule cells (GCs) within the molecular layer. PCs relay the signals from GCs and other interneurons and send the final output to the deep cerebellar nuclei. Mossy fibers and climbing fibers are the major afferents from outside the cerebellum. Interneurons are located in different layers of the cerebellar cortex: stellate cells and basket cells in the molecular layer; PCs and candelabrum cells in the Purkinje cell layer; and GCs, unipolar brush cells, globular cells, and Lugaro cells in the granular layer. For further details and additional references, please refer to [15,16].

**Table 1 ijms-22-01850-t001:** Examples of cerebellar neuronal markers and the methodologies in which the markers were identified.

Methodology	Neuronal Type and Specific Marker	References
Forward and reverse genetics	Purkinje cell: Grid2, Pcp2 Basket cell: Kcna2	[29,30,42,52,53,54,55,56]
Global screening (in situ hybridization)	Purkinje cell: Esrrb, Nr2f2, Foxp2, Foxp4 Granule cell: Zic1, Neurod1, Etv1, Nfia	[74]
Laser-capture microdissection	Stellate cell: GIRK3	[92]
Single-cell transcriptome	Purkinje cell: Adcy1, Itpka, Csdc2 Granule cell: Cntn2, Tubb3	[95]

## Data Availability

Not applicable.

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
