# Peer review of "In Search of Molecular Markers for Cerebellar Neurons"

_ijms, 2021, doi:10.3390/ijms22041850_

Round 1
Reviewer 1 Report
This is an interesting review from Cheung and his colleagues, describing various technical approaches and recent progresses in the search for cell-specific markers for cerebellar neurons.
The review is nicely presented and will be of interest to a wide audience. The quality of the study is good, only minor revisions are needed.
- The resolution of the Figure1needs to be increased. Labels cannot be clearly seen.
It is missing the synaptic connection between mossy fibers and Lugaro cells (LCs).
Moreover, LC dendritic/soma shape and location in the cerebellar circuit are not correct.
Lugaro cells can be divided into two groups: the first group consists of large-sized Lugaro cells characterized by a triangular soma which occupy the deeper GL The second group consists of small-sized LCs marked by a fusiform soma located underneath the PCs layer.
From the opposite extremities of the LC soma, two pairs of thick, horizontal, rarely ramified dendrites emerge, running parallel to the PCL in the parasagittal plane (Prestori et al., 2019).
- In order to clarify the functional role of each neuron involved in the cerebellar circuit, I would add the plus or minus sign (excitatory synapse, inhibitory synapse) to each connection.
3. The unique reference about the cerebellar role in cognitive function is not absolutely sufficient (Depping et al., 2020). It is necessary to add other references. Some examples:
D’angelo E. (2019)
D’Angelo and Casali (2013)
Hoche at al., (2018)
Schmahmann (2019)
Soda et al., (2019)
Giza et al (2010)
- To better understand the text it would be useful to add a table summarizing the main cerebellar markers and the methodological techniques to identify them
- The PC markers Calb1, Fgf7 should be introduced first together with pcp2 e grid2
Author Response
Reviewer #1
This is an interesting review from Cheung and his colleagues, describing various technical approaches and recent progresses in the search for cell-specific markers for cerebellar neurons.
The review is nicely presented and will be of interest to a wide audience. The quality of the study is good, only minor revisions are needed.
The resolution of the Figure1 needs to be increased. Labels cannot be clearly seen.
It is missing the synaptic connection between mossy fibers and Lugaro cells (LCs).
Moreover, LC dendritic/soma shape and location in the cerebellar circuit are not correct.
Lugaro cells can be divided into two groups: the first group consists of large-sized Lugaro cells characterized by a triangular soma which occupy the deeper GL The second group consists of small-sized LCs marked by a fusiform soma located underneath the PCs layer.
From the opposite extremities of the LC soma, two pairs of thick, horizontal, rarely ramified dendrites emerge, running parallel to the PCL in the parasagittal plane (Prestori et al., 2019).
In order to clarify the functional role of each neuron involved in the cerebellar circuit, I would add the plus or minus sign (excitatory synapse, inhibitory synapse) to each connection.
Response:
Thank you very much for your comment.
We found that the low-resolution problem could be due to the restriction of the manuscript template, which can only allow the figure to be displayed in about 2/3 of the total width. The figure would become much clearer when it is displayed with full width.
Meanwhile, we have revised the figure as follows:
- The font size of the text in the figure has been increased.
- The connection between mossy fibers and Lugaro cells has been added.
- Both fusiform-Lugaro cell and globular cell (which was considered as a Lugaro cell-like cell) have been added.
- “plus” and “minus” signs have been added to the major connections.
- Two references have been added to the figure legend so that readers can be referred to.
D’Angelo et al. (2016). Modeling the cerebellar microcircuit: new strategies for a long-standing issue. Frontiers in Cellular Neuroscience 10, 176.
Prestori et al. (2019). Diverse neuron properties and complex network dynamics in the cerebellar cortical inhibitory circuit. Frontier in Molecular Neuroscience 12, 267.
- The unique reference about the cerebellar role in cognitive function is not absolutely sufficient (Depping et al., 2020). It is necessary to add other references. Some examples:
D’angelo E. (2019).
D’Angelo and Casali (2013).
Hoche at al., (2018).
Schmahmann (2019).
Soda et al., (2019).
Giza et al (2010).
Response:
- Thank you very much for suggesting the references regarding the cognitive function of the cerebellum. All the references recommended have been added to the text.
To better understand the text it would be useful to add a table summarizing the main cerebellar markers and the methodological techniques to identify them.
Response:
- Thank you very much for your suggestion. A table listing some examples of cerebellar neuronal markers (Table 1) has been added.
The PC markers Calb1, Fgf7 should be introduced first together with pcp2 e grid2
Response:
A brief introduction of calbindin (Calb1) has been added in the introduction section 1.3 after the introduction of Pcp2 as the following:
“Another PC-specific marker calbindin (Calb1) has been widely adopted in histological studies of the cerebellum since the late 1980s.”. References have also been added accordingly.
For Fgf7, although some studies showed significant down-regulation of Fgf7 in Purkinje cell-defective models (e.g., pcd3J mouse (Rong et al., 2004), Moonwalker mouse (Dulneva et al., 2015), Lhx1/5 double conditional KO mouse (Lui et al., 2017)), we realize that it is less commonly used independently as a PC-specific marker compared with Pcp2 and Calb1. Considering the deletion of Fgf7 will not affect the meaning of the sentence, we have removed Fgf7 from the text.
References:
Rong et al. (2004). Identification of candidate Purkinje cell-specific markers by gene expression profiling in wild-type and pcd3J mice. Molecular Brain Research 132, 128-145
Dulneva et al. (2015). The mutant Moonwalker TRPC3 channel links calcium signaling to lipid metabolism in the developing cerebellum. Human Molecular Genetics 24, 4114-4125.
Lui et al. (2017). Lhx1/5 control dendritogenesis and spine morphogenesis of Purkinje cells via regulation of Espin. Nature Communications 8, 15079
Reviewer 2 Report
Authors reviewed various technical approaches and recent progress in the search for cell-specific genetic markers for cerebellar neurons. To elucidate the exact roles of each type of neurons consisting the cerebellar cortex, authors are interested in identifying the cell-type specific markers which will render cell-specific labeling and functional study by genetic manipulation, especially markers of interneurons. Generally speaking, description is clear and various techniques are introduced concisely. However, authors omitted important previous researches attempting to clarify roles of each cell type in sophisticated cerebellar function using genetic manipulation and physiological/pharmacological techniques. Following studies should be referred and criticized, if necessary, to explain why technologies introduced in this review are necessary. Otherwise, the latter part of this review seems to be just a list of technologies which are not able to identify useful genetic markers of interneurons, so far.
1.Kono et al., Interneuronal NMDA receptors regulate long-term depression and motor learning in the cerebellum. J. Physiol. 597.3 (2019), 903-920.
2. Watanabe et al., Ablation of cerebellar Golgi cells disrupts synaptic integration involving GABA inhibition and NMDA receptor activation in motor coordination. Cell 95.1 (1998) 17-27.
Physiological properties of the unipolar brush cell are relatively well-studied, and collaboration of whole-cell patch-clamp methods and NGS analysis or other technologies introduced in this review seems to be more successful. It is better to refer and discuss about the following paper.
3. Balmer & Trussell. Selective targeting of unipolar brush cell subtypes by cerebellar mossy fibers. eLife (2019) 8:e44964. DOI:https://doi.org/10.7554/eLife.44964.
Lugaro cell-like globular cell, of which cell body shows globular shape, is classified as a separate type from classic, fusiform Lugaro cell, I think, because noradrenaline elicits robust firing only in the globular neuron, and input from PC is quantitatively different between them.
4.Hirono et al., Cerebellar globular cells receive monoaminergic excitation and monosynaptic inhibition from Purkinje cells. PLoS ONE 7(1), (2012), e29663. doi: 10.1371/journal.pone. 0029663
Thus, number of cell-types in the cerebellar cortex would be nine.
Globular neuron is better to be added in Fig.1, and shape of Lugaro cell should be changed to fusiform.
Author Response
Reviewer #2
Authors reviewed various technical approaches and recent progress in the search for cell-specific genetic markers for cerebellar neurons. To elucidate the exact roles of each type of neurons consisting the cerebellar cortex, authors are interested in identifying the cell-type specific markers which will render cell-specific labeling and functional study by genetic manipulation, especially markers of interneurons. Generally speaking, description is clear and various techniques are introduced concisely. However, authors omitted important previous researches attempting to clarify roles of each cell type in sophisticated cerebellar function using genetic manipulation and physiological/pharmacological techniques. Following studies should be referred and criticized, if necessary, to explain why technologies introduced in this review are necessary. Otherwise, the latter part of this review seems to be just a list of technologies which are not able to identify useful genetic markers of interneurons, so far.
1.Kono et al., Interneuronal NMDA receptors regulate long-term depression and motor learning in the cerebellum. J. Physiol. 597.3 (2019), 903-920.
- Watanabe et al., Ablation of cerebellar Golgi cells disrupts synaptic integration involving GABA inhibition and NMDA receptor activation in motor coordination. Cell 95.1 (1998) 17-27.
Response:
Thank you very much for your comment.
We hope that the reviewer would be convinced that the two studies suggested by the reviewers are indeed benefitted from the technologies described in the manuscript.
The first study (Kono et al., 2019) utilizes several Cre-driver mouse lines, including Pcp2-Cre, Gabra6-Cre, and Pvalb-Cre, to generate mice with various cell-specific knockout of Grin1. These gene promoters have to be well-characterized before these transgenic lines can be created. The discovery of these promoters is the consequence of the adoption of technologies mentioned in the manuscript. For example, Pcp2 was one of the cDNA clones identified in the Purkinje cell degeneration (pcd) mouse (Nordquist et al., 1988), which involves forward genetics and global screening (this global screening of cDNA library will be further explained below).
Similarly, the second study (Watanabe et al., 1998) also involves global screening and reverse genetic methodologies that we have discussed in the manuscript. Watanabe et al. utilized a technique called immunotoxin-mediated cell targeting (IMCT) by using a transgene, in which the expression is under the control of mGluR2 promoter, to specifically eliminate Golgi cells (and that is a reverse genetic approach). Noteworthy, before this study can be performed, several prerequisite works have been conducted to demonstrate the predominant expression of mGluR2 in Golgi cells. Essentially, cDNA clones of mGluRs were isolated from rat brain cDNA library (which is the same global screenings of cDNA library approach in Nordquist et al., 1988) and in situ hybridization/immunostaining was performed to examine the expression patterns of these mGluRs (Abe et al. 1992; Neki et al. 1996; Ohishi et al. 1993a, 1993b, 1994, 1995; Shigemoto et al. 1992).
In the 1980s and early 1990s, cDNA library screening is a commonly used approach to uncover genes that are expressed in a particular tissue. This method is to extract mRNA from a particular tissue and then generate a cDNA library. Genes are identified by DNA sequencing, and in situ hybridization (and immunostaining) is performed to examine the gene expression pattern. This method is no longer be used and is replaced by advanced technology like the single-cell transcriptome described in the manuscript, but the concept is principally the same. That is the reason we did not include the cDNA library screening approach in the manuscript.
Overall, we believe that the two studies you suggested further reinforce the importance of the technologies we mentioned in the manuscript. Without the discovery of cell-specific markers/promoters, it was impossible for the transgenic mice production and thus those downstream physiological/pharmacological studies could become infeasible.
References:
Abe et al. (1992). Molecular characterization of a novel metabotropic glutamate receptor mGluR5 coupled to inositol phosphate/Ca2+ signal transduction. Journal of Biological Chemistry 267, 13361-13368.
Neki et al. (1996). Metabotropic glutamate receptors mGluR2 and mGluR5 are expressed in two non-overlapping populations of Golgi cells in the rat cerebellum. Neuroscience 75, 815-826.
Nordquist et al. (1988). cDNA cloning and characterization of three genes uniquely expressed in cerebellum by Purkinje Neurons. Journal of Neuroscience 8, 4780-4789.
Ohishi et al. (1993a). Distribution of the messenger RNA for a metabotropic glutamate receptor, mGluR2, in the central nervous system of the rat. Neuroscience 53, 1009-1018.
Ohishi et al. (1993b) Distribution of the mRNA for a metabotropic glutamate receptor (mGluR3) in the rat brain: an in situ hybridization study. Journal of Comparative Neurology 335, 252-266.
Ohishi et al. (1994). Immunohistochemical localization of metabotropic glutamate receptors, mGluR2 and mGluR3, in rat cerebellar cortex. Neuron 13, 55-66.
Ohishi et al. (1995). Distribution of the mRNAs for L-2-amino-4-phosphonobutyrate-sensitive metabotropic glutamate receptors, mGluR4 and mGluR7, in the rat brain. Journal of Comparative Neurology 360, 555-570.
Shigemoto R., Nakanishi S. and Mizuno N. (1992) Distribution of the mRNA for a metabotropic glutamate receptor (mGluR1) in the central nervous system: an in situ hybridization study in adult and developing rat. Journal of Comparative Neurology 322, 121-135.
Physiological properties of the unipolar brush cell are relatively well-studied, and collaboration of whole-cell patch-clamp methods and NGS analysis or other technologies introduced in this review seems to be more successful. It is better to refer and discuss about the following paper.
- Balmer & Trussell. Selective targeting of unipolar brush cell subtypes by cerebellar mossy fibers. eLife (2019) 8:e44964. DOI:https://doi.org/10.7554/eLife.44964.
Response:
Thank you very much for your comment.
By applying neuronal tracing from the vestibular nucleus, Balmer & Trussell (2019) nicely showed subtypes of unipolar brush cells. Again, this study is also benefitted from the technology mentioned in the manuscript. The Glt25d2 mouse line used in their study is indeed one of the Cre-driver mouse lines generated by the GENSAT project (Gerfen et al., 2013). Despite this, it is expected that there are subtypes of cerebellar neurons existing, for example, zebrinII+ve/-ve Purkinje cells (Sillitoe et al., 2009), ON/OFF subtypes of unipolar brush cells (Balmer & Trussell, 2019), globular cell which was considered as a subtype of Lugaro cells (Hirono et al., 2012). Hence, it would be critical to have detailed molecular signatures of each subtype and single-cell transcriptome approach would be one of the potential methods to find out.
We hope that you would be convinced about the importance of the technologies mentioned in the manuscript and their strong influences on subsequent downstream studies in genetics, physiology, pharmacology, etc.
References:
Balmer & Trussell (2019). Selective targeting of unipolar brush cell subtypes by cerebellar mossy fibers. eLife 8, e44964.
Gerfen et al. (2013). GENSAT BAC Cre-recombinase driver lines to study the functional organization of cerebral cortical and basal ganglia circuit. Neuron 80, 1368-1383.
Hirono et al. (2012). Cerebellar globular cells receive monoaminergic excitation and monosynaptic inhibition from Purkinje cells. PLoS ONE 7, e29663.
Sillitoe et al. (2009).Embryonic origins of ZebrinII parasagittal stripes and establishment of
topographic Purkinje cell projections. Neuroscience 162, 574–588
Lugaro cell-like globular cell, of which cell body shows globular shape, is classified as a separate type from classic, fusiform Lugaro cell, I think, because noradrenaline elicits robust firing only in the globular neuron, and input from PC is quantitatively different between them.
4.Hirono et al., Cerebellar globular cells receive monoaminergic excitation and monosynaptic inhibition from Purkinje cells. PLoS ONE 7(1), (2012), e29663. doi: 10.1371/journal.pone. 0029663
Thus, number of cell-types in the cerebellar cortex would be nine.
Globular neuron is better to be added in Fig.1, and shape of Lugaro cell should be changed to fusiform.
Response:
Thank you very much for your comment.
We have changed the number of cell types in the cerebellar cortex to nine.
As the other reviewer also commented about Figure 1, we have revised the figure as follows:
- The font size of the text in the figure has been increased.
- The connection between mossy fibers and Lugaro cells has been added.
- Both fusiform-Lugaro cell and globular cell have been added.
- “plus” and “minus” signs have been added to the major connections.
- Two references have been added to the figure legend so that readers can be referred to:
D’Angelo et al. (2016). Modeling the cerebellar microcircuit: new strategies for a long-standing issue. Frontiers in Cellular Neuroscience 10, 176.
Prestori et al. (2019). Diverse neuron properties and complex network dynamics in the cerebellar cortical inhibitory circuit. Frontier in Molecular Neuroscience 12, 267.
Round 2
Reviewer 2 Report
OK
Author Response
Thank you for your comment. We have included the traditional cDNA library screening method in the Section 2.3. Global Screening as the following:
“A traditional way was tissue-specific cDNA library screening, in which mRNAs extracted from cerebellum were used for cDNA library preparation for initial screening. Genes were then identified by DNA sequencing and in situ hybridization (and immunohistochemical staining for gene products) was performed to obtain spatiotemperal gene expression pattern. Pcp2, for example, was one of the cDNA clones identified in the Purkinje cell degeneration (pcd) mouse [72].”